# An Inexact Feasible Quantum Interior Point Method for Linearly Constrained Quadratic Optimization

**DOI:** 10.3390/e25020330

**Published:** 2023-02-10

**Authors:** Zeguan Wu, Mohammadhossein Mohammadisiahroudi, Brandon Augustino, Xiu Yang, Tamás Terlaky

**Affiliations:** Department of Industrial and Systems Engineering, Lehigh University, Bethlehem, PA 18015, USA

**Keywords:** quantum computing, interior point method, quadratic optimization, 90C20, 90C51, 81P68

## Abstract

Quantum linear system algorithms (QLSAs) have the potential to speed up algorithms that rely on solving linear systems. Interior point methods (IPMs) yield a fundamental family of polynomial-time algorithms for solving optimization problems. IPMs solve a Newton linear system at each iteration to compute the search direction; thus, QLSAs can potentially speed up IPMs. Due to the noise in contemporary quantum computers, quantum-assisted IPMs (QIPMs) only admit an inexact solution to the Newton linear system. Typically, an inexact search direction leads to an infeasible solution, so, to overcome this, we propose an inexact-feasible QIPM (IF-QIPM) for solving linearly constrained quadratic optimization problems. We also apply the algorithm to $\ell_{1}$-norm soft margin support vector machine (SVM) problems, and demonstrate that our algorithm enjoys a speedup in the dimension over existing approaches. This complexity bound is better than any existing classical or quantum algorithm that produces a classical solution.

## 1. Introduction

Linearly constrained quadratic optimization (LCQO) is defined as optimizing a convex quadratic objective function over a set of linear constraints. Linear optimization is a special case of LCQO that corresponds to the case where the objective function is linear. LCQO has rich theory, algorithms, and applications. Many problems in machine learning can be formulated as LCQO problems, including variants of least square problems and variants of support vector machine training [1,2]. Some important optimization algorithms also have LCQO subproblems, e.g., sequential quadratic programming [1].

The modern age of IPMs was launched by Karmarkar’s projective method for linear optimization (LO). Since then, many variants of IPMs have also been applied to nonlinear optimization problems, including LCQO problems [3,4]. Contemporary IPMs progress towards the set of optimal solutions by moving within a neighbourhood of an analytic curve known as the central path. IPMs can be categorized according to whether or not the the sequence of iterates produced by the algorithm satisfies feasibility. Feasible IPMs are initialized with a strictly feasible solution and maintain feasibility in each iteration, whereas infeasible IPMs start from an infeasible interior solution and do not require feasibility to be exactly satisfied at any point of the algorithm. For LCQO problems with *n* variables, feasible IPMs can produce an ϵ-approximate solution using $O(\sqrt{n}\log\left(1/\epsilon \right))$ iterations, whereas infeasible IPMs require $O(n^{2}\log\left(1/\epsilon \right))$ IPM iterations to converge to an ϵ-approximate solution [5,6].

At each IPM iteration, a linear system needs to be solved to obtain the search direction, called the Newton direction. This so-called Newton linear system is traditionally in the form of the augmented system or the normal equation system. Classically, these linear systems can be solved exactly using Bunch–Parlett factorization if the matrices in the systems are symmetric indefinite [7], or Cholesky factorization if the matrices are symmetric positive definite. Solving the Newton linear systems using direct factorization approaches requires the use of $O(n^{3})$ arithmetic operations, which suggests that feasible IPMs based on factoring methods cannot exhibit complexity better than $O(n^{3.5}\log\left(1/\epsilon \right))$, whereas, with the partial update, they achieve $O(n^{3}\log\left(1/\epsilon \right))$ arithmetic operation complexity. The linear systems can also be solved inexactly using some inexact methods, e.g., Krylov subspace methods, which may require fewer iterations if the desired accuracy of the solutions to the linear systems is not high. However, inaccurately solving the Newton linear systems (i.e., the inaccuracy of the search directions) may result in the infeasibility of the sequence of solutions generated by IPMs; therefore, they have only been used in infeasible IPMs.

The advent of quantum technology has led to the development of many quantum-assisted algorithms for optimization and machine learning applications, such as linear regression [8] and the support vector machine training problem [9]. Following the seminal work on quantum algorithms for solving linear systems of equations [10], researchers have been studying whether QLSAs could yield quantum speedups in classical optimization algorithms. In particular, quantum IPMs (QIPMs) that utilize QLSAs to solve the Newton linear system arising in each iteration have been proposed for LO problems [11,12] and semidefinite optimization problems [13]. To maintain the feasibility of the iterates using quantum subroutines, the authors of [13,14] introduce the so-called orthogonal subspace system (OSS) for SDO and LO problems, and, in particular, demonstrate that a feasible solution to the original Newton system can be recovered from an inexact solution to the OSS. However, linearly constrained quadratic optimization problems, which are fundamental to both optimization and machine learning, have yet to be formally studied in the quantum literature.

In this work, we generalize the OSS for LO problems in [14] to LCQO problems and provide an efficient method for constructing the OSS using a quantum computer. Using the OSS, we can obtain an inexact feasible IPM, solving for the search directions inexactly but maintaining the feasibility of the iterates throughout the process of our IPM. The feasibility of the iterates gives better IPM iteration complexity and the bottleneck becomes solving the linear system, OSS. In particular, we show that a quantum implementation of our algorithm with access to quantum RAM (QRAM) obtains an ϵ-approximate solution to a given LCQO problem with worst-case complexity
$${\tilde{O}}_{n,\bar{\omega },\frac{1}{\epsilon }}\left(\sqrt{n}\left(n\left(\frac{{\bar{\omega }}^{2}}{\epsilon }+\sigma_{\max}\left(Q\right)\right)\kappa_{VAQ}+n^{2}\right)\right),$$
where $\bar{\omega }=\max_{k}\omega^{k}$, $\sigma_{\max}\left(Q\right)$ is the maximum singular value of the Hessian of the objective function and $\kappa_{VAQ}$ is the condition number of a matrix determined by initial data; see Lemma 3. We also consider the application of $\ell_{1}$-norm soft margin SVM problems, in which case, an ϵ-approximate solution is obtained with complexity
$${\tilde{O}}_{m,n,\bar{\omega },\frac{1}{\epsilon }}\left({\left(m+n\right)}^{1.5}\left(\frac{{\bar{\omega }}^{2}}{\epsilon }+\sigma_{\max}\left(Q\right)\right)\kappa_{VAQ}+{\left(m+n\right)}^{2.5}\right).$$

Here, *m* is the number of features and *n* is the number of data points. $\bar{\omega }$, *Q*, and $\kappa_{VAQ}$ are defined similarly from the LCQO formulation of the SVM problem; see Section 4. The dependence on dimension is better than any existing quantum or classical algorithm.

The rest of this paper is organized as follows: in Section 2, we introduce IPMs for LCQO and the OSS system; in Section 3, we discuss how to use quantum algorithms to find the Newton directions and analyze the complexity of our IF-QIPM; in Section 4, we apply our IF-QIPM to the support vector machine problem. Discussions are provided in Section 5, and some technical proofs are moved to the Appendix A and Appendix B.

## 2. Preliminaries

In this section, we introduce notations before reviewing the theory of IPMs applied to LCQO, and derive the OSS system for the class of problems.

### 2.1. Notation

Vectors are typically represented by lower-case letters. We write $0_{n}$ when referring to the *n*-dimensional all-zeros vector, and the *n*-dimensional all-ones vector is denoted by $e_{n}$. When the dimension is obvious from the context, we may write 0 or *e*, respectively. Matrices are typically represented with upper-case letters. The identity of dimension *n* is denoted by $I_{n\times n}$, and $0_{n\times m}$ represents the n×m-dimensional all-zero matrix, again, dropping these subscripts when the dimension is obvious from the context. For a general n×m-dimensional matrix *H*, we write $H_{i\cdot }$ to refer to its *i*th row, and, similarly, denote the *j*th column by $H_{\cdot j}$. For the (i,j)th element of *H*, we write $H_{ij}$ or $H_{i,j}$.

For real-valued functions $f_{1}$, $f_{2}$, and $f_{3}$, we write
$$f_{1}=O\left(f_{2}\right)$$
if there exists a positive number $k_{4}$ such that $f_{1}\leq k_{4}f_{2}$. We write
$$f_{1}={\tilde{O}}_{f_{3}}\left(f_{2}\right)$$
if there exists a positive number $k_{5}$ such that $f_{1}\leq k_{5}f_{2}\times \mathrm{poly}\log\left(f_{3}\right)$.

### 2.2. IPMs for LCQO

In this work, LCQO is defined as follows.

 **Definition 1**(LCQO Problem)**.**
*For vectors $b\in R^{m}$, $c\in R^{n}$, and matrices $A\in R^{m\times n}$ and $Q\in R^{n\times n}$ with rank(A)=m≤n and Q symmetric positive semidefinite, we define the primal and dual LCQO problems as:*
(1)$$\left(P\right)\;\begin{array}{cc} \min\;c^{T} & x+\frac{1}{2}x^{T}Qx,\\ s.t.\;\;Ax & =b,\\ x & \geq 0, \end{array}\;\;\left(D\right)\;\begin{array}{cc} \max\;b^{T}y-\frac{1}{2}x^{T}Q & x,\;\;\\ s.t.\;\;A^{T}y+s-Qx & =c,\\ s & \geq 0, \end{array}$$
*where $x\in R^{n}$ is the vector of primal variables, and $y\in R^{m}$, $s\in R^{n}$ are vectors of the dual variables. Problem (P) is called the primal problem and (D) is called the dual problem.*

Since *A* is of full row-rank, *A* does not contain any null rows, and we further make the following assumption on matrix *A*.

**Assumption 1.** 
*Matrix A has no all-zero columns.*


**Remark 1.** 
*Suppose that A has zero columns. Without a loss of generality, assume that the nth column is all-zero. Introducing a new variable $x_{n+1}$, we can rewrite the problem as*

$$\begin{array}{cc} \min\;{\left[\begin{array}{c} c\\ 0 \end{array}\right]}^{T}\left[\begin{array}{c} x\\ x_{n+1} \end{array}\right]+\frac{1}{2}{\left[\begin{array}{c} x\\ x_{n+1} \end{array}\right]}^{T}\left[\begin{array}{cc} Q & 0_{n\times 1}\\ 0_{1\times n} & 0 \end{array}\right] & \left[\begin{array}{c} x\\ x_{n+1} \end{array}\right],\\ s.t.\;\;\left[\begin{array}{ccccc} A_{\cdot 1} & \cdots & A_{\cdot (n-1)} & 0_{m\times 1} & 0_{m\times 1}\\ 0 & \cdots & 0 & 1 & -1 \end{array}\right]\left[\begin{array}{c} x\\ x_{n+1} \end{array}\right] & =\left[\begin{array}{c} b\\ 0 \end{array}\right],\\ x\geq 0,\;x_{n+1} & \geq 0. \end{array}$$


*The new LCQO problem is equivalent to the original one, and contains fewer all-zero columns. Iterating this procedure to eliminate each of the all-zero columns, we obtain a new LCQO problem satisfying Assumption 1 with no more than 2n−m variables and n constraints in the worst case.*


**Assumption 2.** 
*There exists a solution $\left(x,\;y,\;s\right)\in R^{n}\times R^{m}\times R^{n}$ such that*

$$Ax=b,\;x>0,\;A^{T}y+s-Qx=c,\;\mathrm{and}\;s>0.$$



The set of *primal–dual feasible solutions* is defined as
$$\mathrm{PD}:=\left\{\left(x,\;y,\;s\right)\in R^{n}\times R^{m}\times R^{n}:Ax=b,\;A^{T}y+s-Qx=c,\;\left(x,s\right)\geq 0\right\}$$
and, similarly, the set of *interior feasible primal–dual solutions* is given by
$${\mathrm{PD}}^{0}:=\left\{\left(x,\;y,\;s\right)\in R^{n}\times R^{m}\times R^{n}:Ax=b,\;A^{T}y+s-Qx=c,\;\left(x,s\right)>0\right\}.$$

By strong duality, the set of optimal solutions can be characterized as
$${\mathrm{PD}}^{*}:=\left\{(x,\;y,\;s)\in \mathrm{PD}:xs=0\right\},$$
where xs denotes the Hadamard, i.e., component-wise product of *x* and *s*. Let ϵ>0; then, the set of ϵ-approximate solutions to Problem (1) can be defined as
(2)$${\mathrm{PD}}_{\epsilon }:=\left\{\left(x,\;y,\;s\right)\in \mathrm{PD}:x^{T}s\leq n\epsilon \right\}.$$

Let *X* and *S* be diagonal matrices of *x* and *s*, respectively. Under Assumption 2, for all μ>0, the perturbed system of optimality conditions
(3)$$\begin{array}{cc} Ax & =b,\\ A^{T}y+s-Qx & =c,\\ XSe & =\mu e,\\ \left(x,s\right) & \geq 0 \end{array}$$
has a unique solution (x(μ),y(μ),s(μ)), and this set of solutions gives rise to the primal–dual central path
$$\mathrm{CP}:=\left\{\left(x,y,s\right)\in {\mathrm{PD}}^{0}|x_{i}s_{i}=\mu \;\mathrm{for}\;i\in \left\{1,\cdots ,n\right\};\;\mathrm{for}\;\mu >0\right\}.$$

IPMs apply Newton’s method to solve system (Equation 3). At each iteration of infeasible IPMs, a candidate solution to the primal–dual LCQO pair in (Equation 1) is updated by solving the following linear system to find the Newton direction:(4)$$\left[\begin{array}{ccc} A & 0 & 0\\ -Q & A^{T} & I\\ S & 0 & X \end{array}\right]\left[\begin{array}{c} \Delta x\\ \Delta y\\ \Delta s \end{array}\right]=\left[\begin{array}{c} r_{p}\\ r_{d}\\ r_{c} \end{array}\right],$$
where
$$\begin{array}{cc} r_{p} & =b-Ax\\ r_{d} & =c-A^{T}y-s\\ r_{c} & =\sigma \mu e-XSe, \end{array}$$
are residuals, and σ∈(0,1) is the barrier reduction parameter. If $r_{p}=0$ and $r_{d}=0$, then the solution (x,y,s) exactly satisfies primal–dual feasibility. We can also define residuals in different ways as we will show later. Once the Newton direction is found, one can move along the direction but has to stay in a neighbourhood of the central path, which is defined as
(5)$$N_{2}\left(\theta \right):=\left\{\left(x,y,s\right)\in {\mathrm{PD}}^{0}{|∥XSe-\mu e∥}_{2}\leq \theta \mu \right\},$$
where θ∈(0,1).

Until relatively recently, inexact solution approaches to solve the Newton linear system (Equation 4) had only been utilized in inexact infeasible IPMs (II-IPMs). For LCQO problems, ref. [6] proposes an II-IPM using an iterative method to solve the Newton systems and obtains a worst-case iteration complexity $O(n^{2}\log\left(\frac{1}{\epsilon }\right))$. On the other hand, feasible IPMs for LCQO problems enjoy $O(\sqrt{n}\log\left(\frac{1}{\epsilon }\right))$ iteration complexity [15,16,17]. In [5], the author provides a general inexact feasible IPM for LCQO problems but does not discuss how the sequence of iterates could be guaranteed to maintain primal–dual feasibility exactly when using inexact linear system solvers. This is a vital consideration, as the feasible neighborhood of the central path as outlined in (Equation 5) is a subset of the primal–dual feasible set; if primal and dual feasibility are not satisfied exactly at any point in the algorithm, the iterates leave this neighborhood and the method fails. Our work fills this gap by using a method inspired by the QIPMs of [13,14].

### 2.3. Orthogonal Subspaces System

Assume that $\left(x,\;y,\;s\right)\in {\mathrm{PD}}^{0}$. To maintain the feasibility of the primal and dual variables, the first two linear equations in system (Equation 4) need to be solved with $r_{p}=0$ and $r_{d}=0$ exactly, which can be guaranteed if Δx lies in the null space of *A*, denoted as Null(A), and $\Delta s=Q\Delta x-A^{T}\Delta y$. Accordingly, we can rewrite system (Equation 4) by representing Δx as a linear combination of basis elements of Null(A). To achieve this, we partition *A* as $A=\left[\begin{array}{cc} A_{B} & A_{N} \end{array}\right]$, where $A_{B}$ is a basis of *A*. Then, we construct the following matrix:$$V=\left[\begin{array}{c} A_{B}^{-1}A_{N}\\ -I \end{array}\right].$$

Matrix *V* has a full column rank and satisfies AV=0, i.e., the columns of *V* span the null space of *A*. Let Δx=Vλ, where $\lambda \in R^{n-m}$ is the unknown coefficient vector used to determine Δx. Subsequently, we can rewrite system (Equation 4) by substituting Δx and Δs in the third equation as
(6)$$\begin{array}{c} SV\lambda +X\left(QV\lambda -A^{T}\Delta y\right)=r_{c}\Leftrightarrow \left[\begin{array}{cc} SV+XQV & -XA^{T} \end{array}\right]\cdot \left[\begin{array}{c} \lambda \\ \Delta y \end{array}\right]=r_{c}. \end{array}$$

A similar system was proposed and called “Orthogonal Subspaces System” (OSS) in [13,14], and we use the same name in this work. The matrix in the OSS system (Equation 6) is of size n×n, and it is nonsingular. Even if the OSS system is solved inexactly, primal and dual feasibility are preserved by computing Δx=Vλ and $\Delta s=QV\lambda -A^{T}\Delta y$. Thus, we can conclude that any inexactness will only impact the third equation of (Equation 4), i.e., $r_{p}=0$ and $r_{d}=0$. This property of the OSS system is very convenient when analyzing the proposed inexact IPM, and allows us to obtain the best known iteration complexity for IPMs.

## 3. Inexact Feasible IPM with QLSAs

In this section, we propose our IF-QIPM for LCQO problems. We begin with the IF-IPM structure introduced by [5] and describe how to quantize it into an IF-QIPM. Then, we analyze the construction of the OSS system and conclude by analyzing the overall complexity of our IF-QIPM.

### 3.1. IF-IPM for LCQO

In [5], the author studies a general conceptual form IF-IPM for QCLO problems by assuming the feasibility of the primal and dual iterates, which induces the following system:(7)$$\left[\begin{array}{ccc} A & 0 & 0\\ -Q & A^{T} & I\\ S & 0 & X \end{array}\right]\left[\begin{array}{c} \Delta x\\ \Delta y\\ \Delta s \end{array}\right]=\left[\begin{array}{c} 0\\ 0\\ r_{c} \end{array}\right],$$
where $r_{c}=\sigma \mu e-XSe$, with σ∈(0,1) being the reduction factor of the central path parameter μ, i.e., $\mu^{new}=\sigma \mu$. When system (Equation 7) is solved with $r_{c}=\sigma \mu e-XSe$ inexactly yielding an error *r*, if ${∥r∥}_{2}\leq \delta {∥r_{c}∥}_{2}$ for some δ∈(0,1), the inexact IPM converges to an ϵ-approximate solution to Problem (1) in at most $O(\sqrt{n}\log\left(1/\epsilon \right))$ iterations. As we mentioned earlier, it is not specified in [5] how to preserve primal and dual feasibility when system (Equation 7) is solved inexactly. Thus, it is presently not clear whether one could recover the convergence conditions described in [5] using inexact approaches, which are reliant on the assumption of primal–dual feasibility (see, e.g., system (Equation 7)).

Now, we present a general procedure of how to solve system (Equation 7) inexactly, while the inexactness error occurs only in the third equation of system (Equation 7). Let (λ,Δy) be an inexact solution for system (Equation 6) and *r* be the error at this solution, i.e.,
$$\begin{array}{c} \left[\begin{array}{cc} SV+XQV & -XA^{T} \end{array}\right]\cdot \left[\begin{array}{c} \lambda \\ \Delta y \end{array}\right]=r_{c}+r. \end{array}$$

The corresponding Newton step
$$\begin{array}{cc} \Delta x & =V\lambda \\ \Delta s & =Q\Delta x-A^{T}\Delta y \end{array}$$
satisfies
$$\left[\begin{array}{ccc} A & 0 & 0\\ -Q & A^{T} & I\\ S & 0 & X \end{array}\right]\cdot \left[\begin{array}{c} \Delta x\\ \Delta y\\ \Delta s \end{array}\right]=\left[\begin{array}{c} 0\\ 0\\ r_{c}+r \end{array}\right].$$

Recall that once (λ,Δy) is determined, then (Δx,Δs) is also (uniquely) determined. An interesting property is that, if (λ,Δy) and (Δx,Δy,Δs) can be deduced from each other, then the OSS system and system (Equation 7) yield the same error term *r*. Hence, the convergence conditions built upon system (Equation 7) can be directly examined using the residual $r_{c}$ and error *r* of the OSS system. Let $\epsilon_{OSS}$ be the target accuracy of the OSS system (Equation 6), i.e.,
$$∥\left(\lambda -\lambda^{*},\;\Delta y-\Delta y^{*}\right){∥}_{2}\leq \epsilon_{OSS},$$
where $\left(\lambda^{*},\;\Delta y^{*}\right)$ is the accurate solution. According to [5], in order to guarantee that the IF-IPM converges, we must have
$$\begin{array}{cc} {∥r∥}_{2} & ={∥\left[\begin{array}{cc} SV+XQV & -XA^{T} \end{array}\right]\cdot \left[\begin{array}{c} \lambda \\ \Delta y \end{array}\right]-r_{c}∥}_{2}\\ \; & \leq {∥\left[\begin{array}{cc} SV+XQV & -XA^{T} \end{array}\right]∥}_{2}\epsilon_{OSS}\\ \; & \leq \delta ∥r_{c}{∥}_{2}, \end{array}$$
where δ∈(0,1) is a constant parameter. Therefore, to ensure the convergence of the IF-IPM, it suffices to set
$$\begin{array}{c} \epsilon_{OSS}\leq \delta \frac{∥r_{c}{∥}_{2}}{{∥\left[\begin{array}{cc} SV+XQV & -XA^{T} \end{array}\right]∥}_{2}}. \end{array}$$

The IF-IPM is presented in full detail in Algorithm 1. In each iteration, we build and solve system (Equation 6) classically. We solve system (Equation 6) to the accuracy just introduced above and then compute the feasible Newton step from the inexact solution and take a full Newton step.
**Algorithm 1: **Short-step IF-IPM1:Choose ϵ>0, δ∈(0,1), θ∈(0,1), β∈(0,1) and $\sigma =(1-\frac{\beta }{\sqrt{n}})$.2:k←03:Choose initial feasible interior solution $\left(x^{0},y^{0},s^{0}\right)\in N\left(\theta \right)$4:**while**$\left(x^{k},y^{k},s^{k}\right)\notin {\mathrm{PD}}_{\epsilon }$**do**5:   $\mu^{k}\leftarrow \frac{{\left(x^{k}\right)}^{T}s^{k}}{n}$6:   $\epsilon_{OSS}^{k}\leftarrow \delta {∥r_{c}^{k}∥}_{2}/{∥\left[\begin{array}{cc} S^{k}V+X^{k}QV^{k} & -X^{k}A^{T} \end{array}\right]∥}_{2}$7:   $(\lambda^{k},\Delta y^{k})\leftarrow$**solve** system (Equation 6) with accuracy $\epsilon_{OSS}^{k}$8:   $\Delta x^{k}=V\lambda^{k}$ and $\Delta s^{k}=-A^{T}\Delta y^{k}$9:   $\left(x^{k+1},y^{k+1},s^{k+1}\right)\leftarrow \left(x^{k},y^{k},s^{k}\right)+\left(\Delta x^{k},\Delta y^{k},\Delta s^{k}\right)$10:   k←k+111:**end while**12:**return**$\left(x^{k},y^{k},s^{k}\right)$

In the quantum-assisted IF-IPM, or IF-QIPM, we propose accelerating Step 7 using quantum subroutines. In the next sections, we investigate how to use quantum algorithms to build and solve the OSS system and obtain the Newton direction.

### 3.2. IF-QIPM for LCQO

The pseudocode of our IF-QIPM is presented in Algorithm 2. At each iteration of the IF-QIPM, we construct and solve system (Equation 6) and compute the Newton direction using quantum algorithms. To obtain an $\epsilon_{OSS}$-approximate solution of system (Equation 6), we first block encode system (Equation 8); see Appendix A. Then, we use quantum algorithms to solve for an $\epsilon_{QLSA}$-approximate solution of system (Equation 8). This solution is normalized but we can rescale it to obtain an $\epsilon_{OSS}$-approximate solution of system (Equation 6). Details are discussed later in this section.
**Algorithm 2: **Short-step IF-QIPM1:Choose ϵ>0, δ∈(0,1), $\theta \in (0,\theta_{0})$, β∈(0,1) and $\sigma =(1-\frac{\beta }{\sqrt{n}})$.2:k←03:Choose initial feasible interior solution $\left(x^{0},y^{0},s^{0}\right)\in N\left(\theta \right)$4:**while**$\left(x^{k},y^{k},s^{k}\right)\notin {\mathrm{PD}}_{\epsilon }$**do**5:   $\mu^{k}\leftarrow \frac{{\left(x^{k}\right)}^{T}s^{k}}{n}$6:   $\epsilon_{OSS}^{k}\leftarrow \delta {∥r_{c}^{k}∥}_{2}/\left(\sqrt{2}{∥\left[\begin{array}{cc} S^{k}V+X^{k}QV^{k} & -X^{k}A^{T} \end{array}\right]∥}_{2}\right)$7:   $(\lambda^{k},\Delta y^{k})\leftarrow$**solve** system (Equation 6) with accuracy $\epsilon_{OSS}^{k}$ quantumly8:   $\Delta x^{k}=V\lambda^{k}$ and $\Delta s^{k}=-A^{T}\Delta y^{k}$9:   $\left(x^{k+1},y^{k+1},s^{k+1}\right)\leftarrow \left(x^{k},y^{k},s^{k}\right)+\left(\Delta x^{k},\Delta y^{k},\Delta s^{k}\right)$10:   k←k+111:**end while**12:**return**$\left(x^{k},y^{k},s^{k}\right)$

Here, $\theta_{0}<1$ and its value will be discussed later. First, we introduce some notations to simplify the OSS system. In the *k*th iteration of Algorithm 2, let
$$\begin{array}{c} M^{k}=\left[\begin{array}{cc} S^{k}V+X^{k}QV & -X^{k}A^{T} \end{array}\right],\;z^{k}=\left[\begin{array}{c} \lambda^{k}\\ \Delta y^{k} \end{array}\right]. \end{array}$$

Then, the OSS system can be rewritten as
$$M^{k}z^{k}=r_{c}^{k}.$$

As discussed in [14], to solve the OSS system (Equation 6) using quantum algorithms, we can first rewrite it as the normalized Hermitian OSS system
(8)$$\begin{array}{c} \frac{1}{\sqrt{2}{∥M^{k}∥}_{F}}\left[\begin{array}{cc} 0 & M^{k}\\ {\left(M^{k}\right)}^{T} & 0 \end{array}\right]\cdot \left[\begin{array}{c} 0\\ z^{k} \end{array}\right]=\frac{1}{\sqrt{2}{∥M^{k}∥}_{F}}. \end{array}\left[\begin{array}{c} r_{c}^{k}\\ 0 \end{array}\right].$$

To use the QLSAs mentioned earlier, we need to turn the linear system (Equation 8) into a quantum linear system using the block encoding introduced in [18]. To this end, we first decompose the coefficient matrix in linear system (Equation 8) as
(9)$$\begin{array}{cc} \frac{1}{\sqrt{2}{∥M^{k}∥}_{F}}\left[\begin{array}{cc} 0 & M^{k}\\ {\left(M^{k}\right)}^{T} & 0 \end{array}\right] & =\frac{1}{\sqrt{2}{∥M^{k}∥}_{F}}\left[\begin{array}{cc} 0 & 0\\ {\left(M^{k}\right)}^{T} & 0 \end{array}\right]+\frac{1}{\sqrt{2}{∥M^{k}∥}_{F}}\left[\begin{array}{cc} 0 & M^{k}\\ 0 & 0 \end{array}\right], \end{array}$$
where
(10)$$\begin{array}{cc} \left[\begin{array}{cc} 0 & 0\\ {\left(M^{k}\right)}^{T} & 0 \end{array}\right]= & \left[\begin{array}{ccc} 0_{n\times n} & 0_{n\times n} & 0_{n\times n}\\ 0_{(n-m)\times n} & V^{T} & 0_{(n-m)\times n}\\ 0_{m\times n} & 0_{m\times n} & -A \end{array}\right]\times \\ \; & \left(\left[\begin{array}{cc} 0_{n\times n} & 0_{n\times n}\\ S^{k} & 0_{n\times n}\\ 0_{n\times n} & 0_{n\times n} \end{array}\right]+\left[\begin{array}{ccc} 0_{n\times n} & 0_{n\times n} & 0_{n\times n}\\ 0_{n\times n} & Q & 0_{n\times n}\\ 0_{n\times n} & 0_{n\times n} & I_{n\times n} \end{array}\right]\left[\begin{array}{cc} 0_{n\times n} & 0_{n\times n}\\ X^{k} & 0_{n\times n}\\ X^{k} & 0_{n\times n} \end{array}\right]\right). \end{array}$$

To compute matrix *V*, we need to find a basis matrix $A_{B}$ of matrix *A* and we need to compute the inverse matrix $A_{B}^{-1}$. Both steps are nontrivial and can be expensive. However, we can reformulate the LCQO problem as follows:$$\begin{array}{cc} \min\;c^{T}x & +\frac{1}{2}x^{T}Qx\\ s.t.\;\left[\begin{array}{ccc} I & 0 & A\\ 0 & I & -A \end{array}\right] & \left[\begin{array}{c} x^{\prime }\\ x^{\prime \prime }\\ x \end{array}\right]=\left[\begin{array}{c} b\\ -b \end{array}\right]\\ x\geq 0,\;x^{\prime } & \geq 0,x^{\prime \prime }\geq 0. \end{array}$$

In this case, we have an obvious basis
$$A_{B}=\left[\begin{array}{cc} I & 0\\ 0 & I \end{array}\right]$$
and matrix *V* can be constructed efficiently
$$V=\left[\begin{array}{c} A_{B}^{-1}A_{N}\\ -I \end{array}\right]=\left[\begin{array}{c} \left[\begin{array}{cc} I & 0\\ 0 & I \end{array}\right]\left[\begin{array}{c} A\\ -A \end{array}\right]\\ -I \end{array}\right]=\left[\begin{array}{c} A\\ -A\\ -I \end{array}\right].$$

Since matrix *A* has no all-zero rows, matrix *V* has no all-zero rows either. This property of the reformulation is useful in the analysis of the proposed IF-QIPM but we do not want to build the complexity analysis on the reformulated problem. Thus, without a loss of generality we may make the following assumption.

**Assumption 3.** *Matrix A is of the form*$A=\left[\begin{array}{cc} I & A_{N} \end{array}\right]$.

To simplify the analysis, we further assume that the input data are integers.

**Assumption 4.** *The input data of Problem* (1) *are integers*.

Based on the two assumptions above, we have the following lemma.

**Lemma 1.** 
*Matrix V equals*

$$V=\left[\begin{array}{c} A_{N}\\ -I \end{array}\right]$$

*and*

$$\min_{i=1,\cdots ,n}\{∥V_{i\cdot }{∥}_{2}^{2}\}=\min\{1,\;\min_{i=1,\ldots ,m}∥{\left(A_{N}\right)}_{i\cdot }{∥}_{2}^{2}\}=1,$$

*where $V_{i\cdot }$ and ${\left(A_{N}\right)}_{i\cdot }$ are the ith row of V and $A_{N}$, respectively.*


Now, we are ready to give $\theta_{0}$ in our definition of the central path neighborhood; see (Equation 5). We set
(11)$$\theta_{0}=\min\left\{\frac{1}{3\sqrt{n}},\;\frac{1}{4{∥QVV^{T}∥}_{F}+1}\right\}.$$

We also define $\omega^{k}$ as the maximum of the values of primal variables and dual slack variables in the *k*th iteration.

**Definition 2.** *Let $\left(x^{k},y^{k},s^{k}\right)$ be a candidate solution for Problem* (Equation 1)*; then,*
$$\omega^{k}=\max_{i\in \{1,\cdots ,n\}}\left\{x_{i}^{k},s_{i}^{k}\right\}.$$

As is standard in the literature on quantum algorithms, in this work, we assume access to quantum random access memory (QRAM). Then, Step 7 of Algorithm 2 consists of three parts: (1). use block encoding to build system (Equation 8); (2). use QLSAs to solve system (Equation 8); (3). use quantum tomography algorithms (QTAs) to extract the classical solution. We use the block-encoding methods introduced in [18] to block-encode linear system (Equation 8).

**Proposition 1.** *In the kth iteration of Algorithm 2, using the block-encoding methods introduced in [18] and the decomposition described in Equations (Equation 9) and (Equation 10), a*$$\begin{array}{c} \left(\frac{\sqrt{{∥V∥}_{F}^{2}+{∥A∥}_{F}^{2}}\sqrt{2}\omega^{k}}{∥M^{k}{∥}_{F}}(\sqrt{2}{∥Q∥}_{F}+\sqrt{2}+1),\;O\left(\mathrm{poly}\log\left(n\right)\right),\;\frac{\epsilon_{QLSA}}{\kappa_{M^{k}}^{3}}\right) \end{array}$$*-block-encoding of the matrix in system (Equation 8) can be implemented efficiently and the complexity will be dominated by the complexity of the QLSA step. Here,*$\epsilon_{QLSA}$*is the accuracy required for the QLSA step and*$\kappa_{M^{k}}$*is the condition number of matrix*$M^{k}$.

**Proof.** See Appendix A for proof. □

Provided access to QRAM, the complexity associated with block encoding the OSS system coefficient matrix and preparing a quantum state encoding the right hand side amounts to polylogarithmic overhead. The cost of these steps is therefore negligible when compared with the complexity contributed by QLSAs and QTAs, so we ignore it here. To bound the total complexity contributed by QLSAs and QTAs, we first need to analyze the accuracy of QLSA characterized by $\epsilon_{QLSA}$, the accuracy of QTA characterized by $\epsilon_{QTA}$, and their relationship.

In each iteration, we use a QLSA to solve the block-encoded version of system (Equation 8) and obtain an $\epsilon_{QLSA}$-approximate solution. Then, we use a QTA to extract an $\epsilon_{QTA}$-approximate solution from the quantum machine. In the context of QLSAs and QTAs, if $\tilde{z}$ is an ϵ-approximate solution of *z*, then $\tilde{z}$ satisfies
$${∥\frac{\tilde{z}}{∥\tilde{z}{∥}_{2}}-\frac{z}{{∥z∥}_{2}}∥}_{2}\leq \epsilon$$

Observe that this definition of accuracy differs from the concept of ϵ-approximate solutions defined in (Equation 2).

Similar to [12,13], the QLSA we use is proposed by [19] and the QTA we use is proposed by [20]. Following the argument in Section 2 in [12], we can establish the relationship among $\epsilon_{QLSA}$, $\epsilon_{QTA}$, and $\epsilon_{OSS}^{k}$ as
(12)$$\begin{array}{c} \epsilon_{QLSA}=\epsilon_{QTA}=\frac{1}{2}\cdot \frac{\sqrt{2}{∥M^{k}∥}_{F}}{∥r_{c}^{k}{∥}_{2}}\epsilon_{OSS}^{k}, \end{array}$$
where $\epsilon_{OSS}^{k}$ is defined as the $\ell_{2}$ norm of the residual when solving system (Equation 8) inexactly in the *k*th iteration. This coefficient is also used to rescale the solution. According to [12], we rescale the normalized solution obtained from QLSA and QTA by
$$\frac{∥r_{c}^{k}{∥}_{2}}{\sqrt{2}{∥M^{k}∥}_{F}}$$
to obtain the $\epsilon_{OSS}^{k}$-approximate solution for system (Equation 6). Here, we did not add superscript to $\epsilon_{QLSA}$ and $\epsilon_{QTA}$, and the reason shall be revealed later. Let
$$\left[\begin{array}{c} {\tilde{0}}^{k}\\ {\tilde{z}}^{k} \end{array}\right]$$
be an inexact solution for system (Equation 8) in the *k*th iteration. Then, the norm of residual of system (Equation 8), which is $\epsilon_{OSS}^{k}$, and the norm of residual of system (Equation 6), which is $∥M^{k}{\tilde{z}}^{k}-r_{c}^{k}{∥}_{2}$, satisfies
$$\begin{array}{cc} \epsilon_{OSS}^{k} & ={∥\frac{1}{\sqrt{2}{∥M^{k}∥}_{F}}\left[\begin{array}{cc} 0 & M^{k}\\ {\left(M^{k}\right)}^{T} & 0 \end{array}\right]\left[\begin{array}{c} {\tilde{0}}^{k}\\ {\tilde{z}}^{k} \end{array}\right]-\frac{1}{\sqrt{2}{∥M^{k}∥}_{F}}\left[\begin{array}{c} r_{c}^{k}\\ 0 \end{array}\right]∥}_{2}\\ \; & ={∥\frac{1}{\sqrt{2}{∥M^{k}∥}_{F}}\left[\begin{array}{c} M^{k}{\tilde{z}}^{k}\\ {\left(M^{k}\right)}^{T}{\tilde{0}}^{k} \end{array}\right]-\frac{1}{\sqrt{2}{∥M^{k}∥}_{F}}\left[\begin{array}{c} r_{c}^{k}\\ 0 \end{array}\right]∥}_{2}\\ \; & \geq {∥\frac{1}{\sqrt{2}{∥M^{k}∥}_{F}}M^{k}{\tilde{z}}^{k}-\frac{1}{\sqrt{2}{∥M^{k}∥}_{F}}r_{c}^{k}∥}_{2}\\ \; & \geq \frac{1}{\sqrt{2}{∥M^{k}∥}_{F}}{∥M^{k}{\tilde{z}}^{k}-r_{c}^{k}∥}_{2}. \end{array}$$

Recall that the error arising from the OSS system (Equation 6) is the same as the error in the full Newton system (Equation 7); then, we can directly use the convergence condition in [5], i.e.,
$$∥M^{k}{\tilde{z}}^{k}-r_{c}^{k}{∥}_{2}\leq \delta {∥r_{c}^{k}∥}_{2}.$$

We can require
$$∥M^{k}{\tilde{z}}^{k}-r_{c}^{k}{∥}_{2}\leq \sqrt{2}∥M^{k}{∥}_{F}\epsilon_{OSS}^{k}\leq \delta {∥r_{c}^{k}∥}_{2}$$
and it follows that
$$\epsilon_{OSS}^{k}\leq \frac{\delta ∥r_{c}^{k}{∥}_{2}}{\sqrt{2}{∥M^{k}∥}_{F}}.$$

Then, choosing
$$\epsilon_{OSS}^{k}=\frac{\delta ∥r_{c}^{k}{∥}_{2}}{\sqrt{2}{∥M^{k}∥}_{F}}\;\mathrm{and}\;\epsilon_{QLSA}=\epsilon_{QTA}=\frac{∥M^{k}{∥}_{F}\epsilon_{OSS}^{k}}{\sqrt{2}{∥r_{c}^{k}∥}_{2}}=\frac{\delta }{2}$$
ensures the convergence of the IF-QIPM. The complexities for each step are also available now. Using the QLSA from [19] and QTA from [20], we have the complexity for QLSA and QTA:$$\begin{array}{cc} T_{QLSA} & ={\tilde{O}}_{n,\bar{\omega },\frac{1}{\epsilon }}\left(\kappa_{M^{k}}\frac{\omega^{k}}{∥M^{k}{∥}_{F}}\right),\\ T_{QTA} & ={\tilde{O}}_{n}\left(\frac{n}{\epsilon_{QTA}}T_{QLSA}\right)={\tilde{O}}_{n,\bar{\omega },\frac{1}{\epsilon }}\left(n\kappa_{M^{k}}\frac{\omega^{k}}{∥M^{k}{∥}_{F}}\right). \end{array}$$

Since we have $\epsilon_{QTA}=\frac{\delta }{2}$ and δ∈(0,1) is a constant parameter, we omit $\epsilon_{QTA}$ in the Big-O notation. Note that the complexity of the block-encoding procedure is dominated by that of QLSA and QTA and thus we ignore the complexity contributed by block encoding. In Step 8, the complexity contributed by computing Newton step from OSS solution is $O(n^{2})$. The total complexity for the *k*th iteration of IF-QIPM will be
(13)$$O\left(T_{QTA}+n^{2}\right)={\tilde{O}}_{n,\bar{\omega },\frac{1}{\epsilon }}\left(\frac{n\omega^{k}\kappa_{M^{k}}}{∥M^{k}{∥}_{F}}+n^{2}\right).$$

#### 3.2.1. Bound for $\omega^{k}/{∥M^{k}∥}_{F}$

In this section, all of the quantities that we consider are from the *k*th iteration. For simplicity, we omit the superscript *k* in this section unless we need it. Using the property of trace, we have
$$\begin{array}{cc} {∥M∥}_{F}^{2} & =\mathrm{tr}(M^{T}M)\\ \; & =\mathrm{tr}\left(\left(SV+XQV\right){\left(SV+XQV\right)}^{T}+XA^{T}AX\right)\\ \; & =\mathrm{tr}\left(\left(SV+XQV\right){\left(SV+XQV\right)}^{T}\right)+\mathrm{tr}\left(XA^{T}AX\right)\\ \; & =\mathrm{tr}\left(SVV^{T}S\right)+\mathrm{tr}\left(XQVV^{T}S\right)+\mathrm{tr}\left(SVV^{T}QX\right)+\mathrm{tr}\left(XQVV^{T}QX\right)+\mathrm{tr}\left(XA^{T}AX\right). \end{array}$$

For the non-symmetric term, due to the cyclic invariant property of trace, we have
$$\mathrm{tr}\left(XQVV^{T}S\right)=\mathrm{tr}\left(SXQVV^{T}\right).$$

Recalling the central path neighborhood that we defined in (Equation 5), we define a matrix *E* such that
(14)$$E=\frac{1}{\mu \theta }\left(XS-\mu I\right).$$

It is obvious that *E* is a diagonal matrix and satisfies
$${∥Ee∥}_{2}<1,$$
which leads to
$$\left|\mathrm{tr}\left(E\right)\right|\leq {∥Ee∥}_{1}\leq \sqrt{n}{∥E∥}_{F}=\sqrt{n}{∥Ee∥}_{2}<\sqrt{n}\;\mathrm{and}\;I-E≻0\;\mathrm{and}\;I+E≻0.$$

With this, we can have
$$\begin{array}{cc} \mathrm{tr}\left(XQVV^{T}S\right) & =\mathrm{tr}\left(SXQVV^{T}\right)\\ \; & =\mathrm{tr}\left(\left(\theta \mu E+\mu I\right)QVV^{T}\right)\\ \; & =\mathrm{tr}\left(\theta \mu EQVV^{T}\right)+\mathrm{tr}\left(\mu QVV^{T}\right). \end{array}$$

For the second term, we know that *Q* and $V^{T}QV$ are both positive semidefinite. Thus, we can have
$$\mathrm{tr}\left(QVV^{T}\right)=\mathrm{tr}\left(V^{T}QV\right)\geq 0$$
because of the cyclic invariant property of trace. According to the Cauchy–Schwarz inequality, we have
$$\mathrm{tr}{\left(EQVV^{T}\right)}^{2}\leq {∥E∥}_{F}^{2}{∥QVV^{T}∥}_{F}^{2}.$$

Thus, we have
$$\mathrm{tr}\left(EQVV^{T}\right)\geq -{∥QVV^{T}∥}_{F}.$$

Thus, we have
$$\begin{array}{cc} \mathrm{tr}\left(XQVV^{T}S\right) & =\mathrm{tr}\left(\theta \mu EQVV^{T}\right)+\mathrm{tr}\left(\mu QVV^{T}\right)\\ \; & \geq \mu \left(\mathrm{tr}\left(QVV^{T}\right)-\theta {∥QVV^{T}∥}_{F}\right)\\ \; & \geq -\theta \mu ∥QVV^{T}{∥}_{F}\\ \; & \geq -\frac{\mu }{4}, \end{array}$$
where the last inequality holds due to condition (Equation 11). Thus, we can bound ${∥M∥}_{F}$ by
$$\begin{array}{cc} {∥M∥}_{F}^{2}= & \mathrm{tr}\left(SVV^{T}S\right)+\mathrm{tr}\left(XQVV^{T}S\right)+\mathrm{tr}\left(SVV^{T}QX\right)\\ \; & \;\;\;\;\;+\mathrm{tr}\left(XQVV^{T}QX\right)+\mathrm{tr}\left(XA^{T}AX\right)\\ \geq & \mathrm{tr}\left(SVV^{T}S\right)+\mathrm{tr}\left(XQVV^{T}QX\right)+\mathrm{tr}\left(XA^{T}AX\right)-\frac{\mu }{2}. \end{array}$$

Since $XQVV^{T}QX⪰0$, we have
$$\begin{array}{cc} {∥M∥}_{F}^{2} & \geq \mathrm{tr}\left(SVV^{T}S\right)+\mathrm{tr}\left(XA^{T}AX\right)-\frac{\mu }{2}. \end{array}$$

Since *X* and *S* are both positive diagonal matrices, we have
$$\begin{array}{cc} {∥M∥}_{F}^{2} & \geq \mathrm{tr}\left(SVV^{T}S\right)+\mathrm{tr}\left(XA^{T}AX\right)-\frac{\mu }{2}\\ \; & =\sum_{i}s_{i}^{2}{\left(VV^{T}\right)}_{ii}+\sum_{i}x_{i}^{2}{\left(A^{T}A\right)}_{ii}-\frac{\mu }{2}\\ \; & \geq \omega^{2}-\frac{\mu }{2}. \end{array}$$

As we mentioned in the very beginning of this section, at each iteration, ω is indeed $\omega^{k}$, but the superscript is ignored here. Now, we aim to find a bound for μ so we can further bound ${∥M∥}_{F}^{2}$. Since ω is the upper bound for the magnitude of the primal and dual slack variables, we have
$$\begin{array}{cc} \omega^{2} & \geq x_{i}s_{i}. \end{array}$$

Recall the definition of matrix *E*; see (Equation 14). Thus, we have
$$\begin{array}{cc} \omega^{2} & \geq x_{i}s_{i}=\mu +\theta \mu E_{ii}\geq \mu -\theta \mu =\left(1-\theta \right)\mu . \end{array}$$

Thus,
$$\begin{array}{cc} {∥M∥}_{F}^{2} & \geq \omega^{2}-\frac{\mu }{2}\geq \omega^{2}-\frac{1}{2}\frac{\omega^{2}}{1-\theta }\geq \omega^{2}-\frac{1}{2}\frac{\omega^{2}}{1-1/3}=\frac{\omega^{2}}{4}, \end{array}$$
where the last inequality follows from the bound for θ; see (Equation 11). Thus, we have
$$\begin{array}{c} \frac{\omega }{{∥M∥}_{F}}\leq 2=O\left(1\right). \end{array}$$

#### 3.2.2. Bound for $\kappa_{M^{k}}$

Similar to the previous section, we ignore the superscript *k* unless we need it. We will start with a general result and then work on the matrix $M^{k}$. The following lemma is a well-known result regarding condition numbers of matrices and can be proven using Courant–Fischer–Weyl min-max principle [21].

**Lemma 2.** 
*For any full row rank matrix $P\in R^{m\times n}$ and symmetric positive definite matrix $D\in R^{n\times n}$, their condition number satisfies*

$$\kappa \left(PDP^{T}\right)\leq \kappa \left(D\right)\kappa \left(PP^{T}\right).$$



Next, we analyze the matrix in the OSS system (Equation 8). Specifically, we focus on $M^{T}M$ since we are interested in the spectral property of the OSS system (Equation 8). Using the matrix *E* defined in (Equation 14), we have the following decomposition:$$\begin{array}{cc} M^{T}M & =\left[\begin{array}{cc} V^{T}{\left(S+XQ\right)}^{T}\left(S+XQ\right)V & -V^{T}{\left(S+XQ\right)}^{T}XA^{T}\\ -AX(S+XQ)V & AX^{2}A^{T} \end{array}\right]\\ \; & =\left[\begin{array}{cc} V^{T}{\left(S+XQ\right)}^{T}\left(S+XQ\right)V & -V^{T}\mu \left(\theta E\right)A^{T}-V^{T}Q^{T}X^{2}A^{T}\\ -A\mu \left(\theta E\right)V-AX^{2}QV & AX^{2}A^{T} \end{array}\right]\\ \; & =\left[\begin{array}{cc} V^{T} & 0\\ 0 & A \end{array}\right]\left[\begin{array}{cc} {\left(S+XQ\right)}^{T}\left(S+XQ\right) & -\mu \theta E-QX^{2}\\ -\mu \theta E-X^{2}Q & X^{2} \end{array}\right]{\left[\begin{array}{cc} V^{T} & 0\\ 0 & A \end{array}\right]}^{T}. \end{array}$$

The second equality holds because
$$\begin{array}{cc} -V^{T}SXA^{T}-V^{T}Q^{T}X^{2}A^{T} & =-V^{T}\mu \left(I+\theta E\right)A^{T}-V^{T}Q^{T}X^{2}A^{T}\\ \; & =-V^{T}\mu \left(\theta E\right)A^{T}-V^{T}QX^{2}A^{T}, \end{array}$$
as AV=0 and *Q* is symmetric. Then, plugging (Equation 14) into the first diagonal block of the decomposition we obtained earlier, we have
$$\begin{array}{cc} M^{T}M= & \left[\begin{array}{cc} V^{T} & 0\\ 0 & A \end{array}\right]\left(\left[\begin{array}{cc} S^{2}+2\mu Q+\mu \theta \left(EQ+QE\right)+QX^{2}Q & -\mu \theta E-QX^{2}\\ -\mu \theta E-X^{2}Q & X^{2} \end{array}\right]\right){\left[\begin{array}{cc} V^{T} & 0\\ 0 & A \end{array}\right]}^{T}\\ = & \left[\begin{array}{cc} V^{T} & 0\\ 0 & A \end{array}\right]\left(\left[\begin{array}{cc} S^{2}+2\mu Q+\mu \theta \left(EQ+QE\right) & -\mu \theta E\\ -\mu \theta E & 0 \end{array}\right]+\left[\begin{array}{cc} QX^{2}Q & -QX^{2}\\ -X^{2}Q & X^{2} \end{array}\right]\right){\left[\begin{array}{cc} V^{T} & 0\\ 0 & A \end{array}\right]}^{T}\\ = & \left[\begin{array}{cc} V^{T} & 0\\ 0 & A \end{array}\right]\left(\left[\begin{array}{cc} I & -Q\\ 0 & I \end{array}\right]\left[\begin{array}{cc} S^{2}+2\mu Q & -\mu \theta E\\ -\mu \theta E & 0 \end{array}\right]\left[\begin{array}{cc} I & 0\\ -Q & I \end{array}\right]\right){\left[\begin{array}{cc} V^{T} & 0\\ 0 & A \end{array}\right]}^{T}\\ \; & +\left[\begin{array}{cc} V^{T} & 0\\ 0 & A \end{array}\right]\left(\left[\begin{array}{cc} I & -Q\\ 0 & I \end{array}\right]\left[\begin{array}{cc} 0 & 0\\ 0 & X^{2} \end{array}\right]\left[\begin{array}{cc} I & 0\\ -Q & I \end{array}\right]\right){\left[\begin{array}{cc} V^{T} & 0\\ 0 & A \end{array}\right]}^{T}\\ = & \left[\begin{array}{cc} V^{T} & 0\\ 0 & A \end{array}\right]\left[\begin{array}{cc} I & -Q\\ 0 & I \end{array}\right]\left[\begin{array}{cc} S^{2}+2\mu Q & -\mu \theta E\\ -\mu \theta E & X^{2} \end{array}\right]\left[\begin{array}{cc} I & 0\\ -Q & I \end{array}\right]{\left[\begin{array}{cc} V^{T} & 0\\ 0 & A \end{array}\right]}^{T}. \end{array}$$

The first two matrices are nonsingular, so we can apply the Lemma 2, and thus we only need to study the middle matrix. Denote the middle matrix by Ψ. Observe that Ψ is almost the same as its counterpart in [14]. Subsequently, we have the following result regarding the spectral property of $M^{k}$.

**Lemma 3.** *When (x,y,s)∈N(θ) and $\theta \in \left(0,\;\min\left\{\frac{1}{3\sqrt{n}},\;\frac{1}{4∥QVV^{T}{∥}_{F}+1}\right\}\right)$, the condition number of matrix $M^{k}$ satisfies*$$\kappa_{M^{k}}=O\left(\frac{{\left(\omega^{k}\right)}^{2}+\mu^{k}\sigma_{\max}\left(Q\right)}{\mu^{k}}\kappa_{VAQ}\right),$$*where*$\kappa_{VAQ}$*is the condition number of the matrix*$\left[\begin{array}{cc} V^{T} & 0\\ 0 & A \end{array}\right]\left[\begin{array}{cc} I & -Q\\ 0 & I \end{array}\right]$.

**Proof.** The proof is in Appendix B. □

Putting all of these together, we have the complexity for our IF-QIPM for LCQO problems.

**Theorem 1.** *The IF-QIPM for LCQO problems stops with the final duality gap less than ϵ in at most*$O\left(\sqrt{n}\log\left(1/\epsilon \right)\right)$*IPM iterations and, in each IPM iteration, the Newton direction can be obtained with complexity*${\tilde{O}}_{n,\bar{\omega },\frac{1}{\epsilon }}\left(n\left(\frac{{\bar{\omega }}^{2}}{\epsilon }+\sigma_{\max}\left(Q\right)\right)\kappa_{VAQ}+n^{2}\right)$*, where*$\bar{\omega }=\max_{k}\omega^{k}$.

**Proof.** The complexity bound for the IPM iterations comes from the result in [5]. According to (Equation 13), the complexity for obtaining the Newton direction is
$$\begin{array}{c} {\tilde{O}}_{n,\bar{\omega },\frac{1}{\epsilon }}\left(\frac{n\omega^{k}\kappa_{M^{k}}}{∥M^{k}{∥}_{F}}+n^{2}\right). \end{array}$$Combining this with the result in Section 3.2.1, the bound in Lemma 3, and $\mu^{k}\geq \epsilon$, we have
$$\begin{array}{c} {\tilde{O}}_{n,\bar{\omega },\frac{1}{\epsilon }}\left(\frac{n\omega^{k}\kappa_{M^{k}}}{∥M^{k}{∥}_{F}}+n^{2}\right)={\tilde{O}}_{n,\bar{\omega },\frac{1}{\epsilon }}\left(n\left(\frac{{\bar{\omega }}^{2}}{\epsilon }+\sigma_{\max}\left(Q\right)\right)\kappa_{VAQ}+n^{2}\right). \end{array}$$□

## 4. Application in Support Vector Machine Problems

In this section, we discuss how to use our IF-QIPM to solve SVM problems. We show that our algorithm can solve $\ell_{1}$-norm soft margin SVM problems faster than any existing classical or quantum algorithms with respect to dimension.

The ordinary SVM problem works on a linearly separable dataset, in which the data points have binary labels. The ordinary SVM aims to find a hyperplane correctly separating the data points with a maximum margin. However, in practice, the data points are not necessarily linearly separable. To allow for mislabelling, the concept of a soft margin SVM was introduced in [22]. Let $\left\{\left(\phi_{i},\zeta_{i}\right)\in R^{m}\times \left\{-1,+1\right\}|i=1,\ldots ,n\right\}$ be the set of data points, Φ be a matrix with the *i*th column being $\phi_{i}$, and *Z* be a diagonal matrix with the *i*th diagonal element being $\zeta_{i}$. The SVM problem with an $l_{1}$-norm soft margin can be formulated as below.
(15)$$\begin{array}{cc} \min_{\left(\xi ,w,t\right)\in R^{n}\times R^{m}\times R}\; & \frac{1}{2}{∥w∥}_{2}^{2}+C{∥\xi ∥}_{1}\\ s.t.\; & \zeta_{i}\left(\langle w,\phi_{i}\rangle +t\right)\geq 1-\xi_{i},\;i=1,\ldots ,n\\ \; & \xi_{i}\geq 0,\;i=1,\ldots ,n. \end{array}$$

Here, (w,t) determines a hyperplane and *C* is a penalty parameter. In [9], the authors rewrote the SVM problem as a second-order conic optimization (SOCO) problem and used the quantum algorithm that they proposed to solve the resulting SOCO problem. They claim the complexity of their algorithm has $O(n^{2})$ dependence on the dimension, which is better than any classical algorithm. However, the algorithm in [9] is invalid. Their algorithm is an inexact infeasible-QIPM (II-QIPM), while they used the IPM complexity for the feasible-QIPM, which ignores at least $O(n^{1.5})$ dependence on *n*. They also missed the symmetrization of the Newton step, which is necessary for SOCO problems and makes their Newton step invalid.

Aside from [9], some pure quantum algorithms for SVM problems are also proposed. In [23], the authors propose a pure quantum algorithm for SVM problems. They claim the complexity is $O(\kappa_{\mathrm{eff}}^{3}\epsilon^{-3}\log\left(mn\right))$, where $\kappa_{\mathrm{eff}}$ is the condition number of a matrix involving the kernel matrix and ϵ is the accuracy. In the worst case, $\kappa_{\mathrm{eff}}=O\left(m\right)$. Their complexity is worse than ours regarding the dependence of dimension and accuracy. In addition, their algorithm does not provide classical solutions. Namely, the solution is in the quantum machine and we cannot read or use it in a classical computer. However, our algorithm produces a classical solution.

To convert the problem into standard-form LCQO, we introduce $\left(w_{+},w_{-}\right)\in R_{+}^{m}\times R_{+}^{m}$, $\left(t_{+},t_{-}\right)\in R_{+}\times R_{+}$, and a slack variable $\rho \in R_{+}^{n}$. Then, we can obtain the following formulation:$$\begin{array}{cc} \min_{w_{+},w_{-},t_{+},t_{-},\xi ,\rho }\; & \frac{1}{2}∥w_{+}-w_{-}{∥}_{2}^{2}+C{∥\xi ∥}_{1}\\ s.t.\; & \zeta_{i}\left(\langle w_{+}-w_{-},\phi_{i}\rangle +t_{+}-t_{-}\right)+\xi_{i}-\rho_{i}=1,\;i=1,\ldots ,n\\ \; & (\xi ,w_{+},w_{-},t_{+},t_{-},\rho )\geq 0. \end{array}$$

This is a standard-form LCQO problem with non-negative variables $(w_{+},w_{-},t_{+},t_{-},\xi ,$$\rho )\in R^{m}\times R^{m}\times R\times R\times R^{n}\times R^{n}$ and parameters
$$\begin{array}{cc} c & =\left[\begin{array}{c} 0_{2m+2}\\ Ce_{n}\\ 0_{n} \end{array}\right]\\ Q & =\left[\begin{array}{ccc} I_{m\times m} & -I_{m\times m} & 0_{m\times (2+2n)}\\ -I_{m\times m} & I_{m\times m} & 0_{m\times (2+2n)}\\ 0_{(2+2n)\times m} & 0_{(2+2n)\times m} & 0_{\left(2+2n)\times (2+2n\right)} \end{array}\right]\\ A & =\left[\begin{array}{cccccc} Z\Phi^{T} & -Z\Phi^{T} & Z & -Z & I_{n\times n} & -I_{n\times n} \end{array}\right]\\ b & =e. \end{array}$$

Thus, we can use the proposed IF-QIPM for LCQO problems to solve the $\ell_{1}$-norm soft margin SVM problems and obtain an ϵ-approximate solution with complexity
$${\tilde{O}}_{m,n,\bar{\omega },\frac{1}{\epsilon }}\left({\left(m+n\right)}^{1.5}\left(\frac{{\bar{\omega }}^{2}}{\epsilon }+\sigma_{\max}\left(Q\right)\right)\kappa_{VAQ}+{\left(m+n\right)}^{2.5}\right).$$

This dependence on dimension is better than any existing quantum or classical algorithm.

## 5. Discussion

In this work, we present an IF-QIPM for LCQO problems by combining the IF-IPM framework proposed in [5] and the OSS system introduced in [14]. Our algorithm has $n^{1.5}$ dependence on *n*, which is better than any existing algorithms for LCQO problems. The dependence on the accuracy is polynomial, which is worse than classic IPMs. Iterative refinement techniques might help to improve the dependence on the accuracy but they are beyond the discussion of this work.

## Data Availability

Not applicable.

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
