# Peer review of "An Inexact Feasible Quantum Interior Point Method for Linearly Constrained Quadratic Optimization"

_entropy, 2023, doi:10.3390/e25020330_

Round 1
Reviewer 1 Report
This paper gives a quantum inexact method for linearly constrained quadratic optimization. It further develops ideas in [5] and [8] to provide the current best quantum method for this problem. The complexity still needs to beat the classical approach. The paper is well-motivated and very well-written.
Author Response
Thank you for your attention and comments! We are still working on beating the classical approach and hopefully it would be successful in the future.
Reviewer 2 Report
In the manuscript, the authors propose an Inexact-Feasible quantum-assisted Interior Point Methods to solve a linearly constrained quadratic optimization problems where both the superiority of quantum linear system algorithms and Interior Point Methods are compatible. The feasible complexity also provides an advantage on the application in the machine learning problems. The analysis about the complexity and application conditions is sufficient. Thus, I recommend it publish in Entropy after the authors considering my tiny question that there is lacking a specific short-steps summary for the step 7 of Algorithm 2. I hope there is instructional short-steps for the Inexact Feasible IPM with QLSAs or at least a visual example to instantiate the advantage of this method.
Author Response
Thank you for your attention and comments! We did not provide a summary for algorithms, which made our work less readable. We have added summaries for both Algorithm 1 and Algorithm 2. Hope these summaries will address your concern. You can find the summary for Algorithm 1 from line 128 and the summary for Algorithm 2 from line 137. The red text below Eq.(12) might also be helpful.